# Hybrid Perovskites Depth Profiling with Variable-Size Argon Clusters and Monatomic Ions Beams

**DOI:** 10.3390/ma12050726

**Published:** 2019-03-02

**Authors:** Céline Noël, Sara Pescetelli, Antonio Agresti, Alexis Franquet, Valentina Spampinato, Alexandre Felten, Aldo di Carlo, Laurent Houssiau, Yan Busby

**Affiliations:** 1Laboratoire Interdisciplinaire de Spectroscopie Electronique, Namur Institute of Structured Matter, University of Namur, 5000 Namur, Belgium; celine.noel@unamur.be (C.N.); laurent.houssiau@unamur.be (L.H.); 2C.H.O.S.E.—Centre for Hybrid and Organic Solar Energy, Department of Electronic Engineering, University of Rome Tor Vergata, 00133 Rome, Italy; pescetel@uniroma2.it (S.P.); antonio.agresti@uniroma2.it (A.A.); aldo.dicarlo@uniroma2.it (A.d.C.); 3IMEC, 3000 Leuven, Belgium; Alexis.Franquet@imec.be (A.F.); Valentina.Spampinato@imec.be (V.S.); 4SIAM platform, University of Namur, 5000 Namur, Belgium; alexandre.felten@unamur.be

**Keywords:** depth profiling, Perovskite solar cells, Argon GCIB, XPS, ToF-SIMS, Low-energy Cesium, hybrid materials

## Abstract

Ion beam depth profiling is increasingly used to investigate layers and interfaces in complex multilayered devices, including solar cells. This approach is particularly challenging on hybrid perovskite layers and perovskite solar cells because of the presence of organic/inorganic interfaces requiring the fine optimization of the sputtering beam conditions. The ion beam sputtering must ensure a viable sputtering rate on hard inorganic materials while limiting the chemical (fragmentation), compositional (preferential sputtering) or topographical (roughening and intermixing) modifications on soft organic layers. In this work, model (Cs_x_(MA_0.17_FA_0.83_)_100−x_Pb(I_0.83_Br_0.17_)_3_/cTiO_2_/Glass) samples and full mesoscopic perovskite solar cells are profiled using low-energy (500 and 1000 eV) monatomic beams (Ar^+^ and Cs^+^) and variable-size argon clusters (Ar_n_^+^, 75 < n < 4000) with energy up to 20 keV. The ion beam conditions are optimized by systematically comparing the sputtering rates and the surface modifications associated with each sputtering beam. X-ray photoelectron spectroscopy, time-of-flight secondary ion mass spectrometry, and in-situ scanning probe microscopy are combined to characterize the interfaces and evidence sputtering-related artifacts. Within monatomic beams, 500 eV Cs^+^ results in the most intense and stable ToF-SIMS molecular profiles, almost material-independent sputtering rates and sharp interfaces. Large argon clusters (n > 500) with insufficient energy (E < 10 keV) result in the preferential sputtering of organic molecules and are highly ineffective to sputter small metal clusters (Pb and Au), which tend to artificially accumulate during the depth profile. This is not the case for the optimized cluster ions having a few hundred argon atoms (300 < *n* < 500) and an energy-per-atom value of at least 20 eV. In these conditions, we obtain (*i*) the low fragmentation of organic molecules, (*ii*) convenient erosion rates on soft and hard layers (but still different), and (*iii*) constant molecular profiles in the perovskite layer, i.e., no accumulation of damages.

## 1. Introduction

Most modern electronic and photonic devices are made of complex hybrid (organic, inorganic) thin film stacks in which ion migrations and interface effects are known to play a dominant role on the device performance and stability. This is particularly true for hybrid perovskite solar cells (PSCs) which are characterized by increasingly sophisticated chemical composition. The chemical engineering of hybrid perovskites has brought complex formulations involving multiple cations (formamidinium, methylammonium, cesium, [1] rubidium [2] and potassium [3]) and mixed anions (iodine and bromine) which have contributed to the spectacular rise of PSCs power conversion efficiency (PCE), which now exceeds 23.5% [3,4,5]. Moreover, state-of-the-art PSCs have demonstrated a considerably improved stability as compared to perovskites based on CH_3_NH_3_PbI_3_ formula. In particular, hybrid perovskite absorbers with formula (FA_x_Cs_1−x_PbI_3_)_0.85_(MAPbBr_3_)_0.15_ have reached an efficiency exceeding 22% (PCE) [5] and prolonged stability thanks to the presence of inorganic atomic cations (Cs^+^) which are believed to promote the uniform growth of large monolithic perovskite grains showing a considerably lower dependence from the processing conditions [1]. 

Despite the undeniable progress in the device stability (above 10000 h with 2D/3D perovskites [6] or with CuSCN/rGO/Au counter electrode [7]), the intrinsic and extrinsic aging mechanisms in hybrid perovskites are still insufficiently understood, and structure-to-performance studies in operated solar cells are rare. Recently, depth profile analyses combining X-Ray photoelectron spectroscopy (XPS) and time of flight secondary ion mass spectrometry (ToF-SIMS) have been successfully applied to investigate aging and failure mechanisms in organic solar cells [8], PSCs [9], and perovskite-based mesoscopic Light-Emitting Diodes (LEDs) [10]. Aging effects in solar cells were investigated by comparing pristine and operated cells after prolonged exposure to light, humidity or oxygen [9,11]. While XPS provides a quantitative chemical surface analysis with moderate sensitivity (accuracy of 1 atomic% and a detection limit of typically 0.1 at%), the ToF-SIMS analysis provides for a semi-quantitative 3D molecular analysis with higher lateral resolution (1 µm vs. a few hundred µm in XPS) and a more accurate depth resolution (1 nm) and detection limit (~ppm). 

For hybrid perovskites, depth profile analysis was applied to investigate the diffusion of atomic and molecular species, the thin layers composition vs. depth, and the perovskite back-conversion into lead iodide or other chemical modifications of layers and interfaces. The objectives of these studies are to rationally optimize the processing conditions and methodology and to investigate interface engineering or aging mechanisms [12,13]. The main limitation of depth profile analysis is related to the ion beam sputtering process, during which energetic impinging ions can introduce artifacts such as the diffusion of small ions by surface charging, [14] intermixing, reduction of metal oxides by preferential sputtering and fragmentation of organic molecules or surface roughening. While ion beam induced modifications cannot be fully avoided, the nature of the sputtering (and analysis) ions and their energy and fluence should be limited to prevent artifacts coming from the accumulation of damages during the profile [15,16,17].

Gas cluster ion beams (GCIBs) and in particular Ar_n_^+^ clusters, with size (*n*) typically ranging from few hundred to few thousand atoms, are the preferred choice to achieve high erosion rates and preserve the molecular composition of soft materials [18]. The advantage of GCIBs sputtering relies on the possible combination of high-energy clusters (*E* up to 40 keV) and low energy-per-atom values (*E/n* ~ few eV). The cluster collision is followed by its disaggregation and the release of its energy to the material surface. Molecular dynamics simulations on cluster collisions have shown that, despite the high sputtering rates, the impact of large argon clusters (*n* > 1000) on organic layers is associated with a weak penetration of Ar atoms, i.e., a restrained in-depth damaging of the material [19,20].

Meanwhile, GCIBs are well-known to be inefficient in sputtering inorganic layers, at least for energies below 20 keV. This feature has considerably limited their applicability in profiling hybrid stacks. Argon GCIBs resulted in ripples formation on silicon [21,22] and gold [23] surfaces and in the reduction of InAs [24], as shown by XPS analysis. A possible strategy to profile inorganic materials with GCIBs is to increase the *E/n* value by reducing the cluster size, however, this has been relatively poorly investigated until now, and guidelines are still controversial [22,25,26]. On the one hand, by lowering the cluster energy (*E* or *E/n*), one would expect to reduce the ion beam induced mixing and thus improve the depth resolution. On the other hand, lowering the cluster energy would result in the lowering of the sputtering rate (profile duration), possibly resulting in the roughening and/or in the accumulation of damages during the profile (induced by both the analysis and the sputtering beam) [27,28,29,30]. For model hybrid samples (*Irganox*) and biological samples, increasing the cluster energy up to 40 keV was shown as beneficial to improve the molecular signals intensities and the lateral resolution [31].

Alternatively, hybrid materials can be sputtered using low-energy (<1 keV) monatomic ion beams (Ar^+^, Cs^+^, O_2_^+^, etc.). In particular, a low-energy Cs^+^ beam has been widely applied in ToF-SIMS profiles of hybrid stacks thanks to (*i*) its ability to sputter organic and inorganic materials at similar rates by the combined chemical-mechanical sputtering, (*ii*) the ability of implanted Cs atoms to increase the negative ionization yield of molecular species, and (*iii*) the ability of Cs atoms to react with radicals formed during the sputtering and limit chemical modifications or crosslinking effects [32]. In this context, for intrinsically hybrid perovskite layers, or for PSCs stacks, combining inorganic (metal and conductive oxides contacts), intrinsically hybrid, and organic layers (as charge carriers extraction layers), it is particularly complicated to predict the best depth profile conditions. Furthermore, systematic studies comparing the surface modifications induced by GCIBs and low-energy monatomic beams are rare.

In this work, we profile hybrid samples based on state-of-the-art triple cation perovskites with nominal composition (FA_x_Cs_1−x_PbI_3_)_0.85_(MAPbBr_3_)_0.15_ deposited on (flat, compact) c-TiO_2_ (referred as model samples) and full mesoscopic solar cells with the stack composition shown in Figure 1. The ion sputtering conditions are varied from low-energy monatomic beams (Ar^+^ and Cs^+^) to variable-size Ar_n_^+^ clusters (100 < n < 4000) ions in the wide energy-per-atom range (from a few eV–100 eV). The methodology consists of combining ToF-SIMS and XPS depth profiles to identify the ion beam induced modifications on the model hybrid samples, and successively, the best conditions are applied to profile the mesoscopic PSCs. Results show that low-energy Cs^+^ sputtering allows the user to quickly profile hybrid materials without inducing the accumulation of modifications, while for argon clusters, the most efficient sputtering conditions correspond to a few hundred atoms clusters with energy-per-atom values above 20 eV. Below this energy, the direct comparison between the depth profiles shows that argon GCIBs are inefficient to sputter metal aggregates (Au, Pb), which leads to their artificial accumulation during the profile.

## 2. Materials and Methods

### 2.1. The Fabrication of Model Perovskite/TiO2 Samples

Patterned fluorine-doped tin oxide (FTO) coated glasses were cleaned in an ultrasonic bath with acetone and 2-propanol, then a compact TiO_2_ (cTiO_2_) blocking layer was deposited by spray pyrolysis from a solution of acetylacetone (2 mL), titanium diisopropoxide (3 mL), and ethanol (45 mL) at 460 °C. The triple cation perovskite layer Cs_x_(MA_0.17_FA_0.83_)_100−x_Pb(I_0.83_Br_0.17_)_3_ was deposited on c-TiO_2_ following the one-step antisolvent deposition method in a nitrogen-filled glovebox. Organic cations were purchased from Dyesol, lead compounds were purchased from TCI Chemicals, and CsI from abcr GmbH (Karlsruhe, Germany). The precursor perovskite solution was prepared by dissolving, with the molar ratio suggested in Reference [1], the mixture of lead iodide (PbI_2_), lead bromide (PbBr_2_), methylammonium bromide (MABr), formamidinium iodide (FAI) and cesium iodide (CsI) in a solvent mixture of anhydrous N,N-dimethylformamide (DMF) and dimethylsulfoxide (DMSO) in a 3:1 volume ratio. The perovskite precursor solution was spin-coated on the TiO_2_ substrate in a two-step program at 1000 and 5000 rpm for 10 and 30 s, respectively. During the second step, 200 μL of chlorobenzene was poured on the spinning substrate 7 s prior to the end of the program. Immediately after spin coating the substrates were annealed at 100 °C for 1 h in a nitrogen-filled glovebox. Glass-sealed samples were sent for the analysis.

### 2.2. Fabrication of Mesoscopic Perovskite Solar Cells 

Glass substrates coated with the patterned FTO and the compact TiO_2_ layer were coated with a thin mesoporous TiO_2_ (mTiO_2)_ film (~150 nm) by spin coating a TiO_2_ paste (Dyesol 30 NR-D paste diluted in ethanol 1:5 in wt.) at 3000 rpm for 20 s, followed by sintering at 460 °C for 30 min in air. The perovskite layer was then deposited on the mesoscopic TiO_2_ as described in the model samples. Then, the spiro-OMeTAD (73.5 g·L^−1^ in chlorobenzene solution doped with TBP (26.7 μL·mL^−1^)), LiTFSI (16.6 μL·mL^−1^), and a Cobalt(III) FK209 complex (7.2 μL·mL^−1^) were sequentially deposited by spin coating at 2000 rpm for 20 s in a glovebox system. Finally, the ~100 nm thick gold counter electrode was deposited by the high-vacuum thermal evaporation through a shadow mask defining an active area of 0.1 cm^2^. The solar cells were encapsulated with a glass lid before being sent for analysis.

### 2.3. XPS Depth Profile Analysis

The XPS depth profile analysis was performed on an ESCALAB 250Xi spectrometer by Thermo Scientific equipped with monatomic Ar^+^ and Ar_n_^+^ GCIB source (MAGCIS) allowing to select the cluster size from *n* = 75 to *n* = 2000 and the cluster energy up to 8 keV. Because of the rapid sputtering rate decay, the cluster energy was not lowered below 6 keV. In particular, the systematic study presented in this work refers to the maximum available energy of 8 keV. Typically, the monatomic beam intensity is the few μA range while for clusters it is ~10 nA. Solar cells were stored in dark conditions under vacuum before the analysis and were analyzed by alternating ion sputtering and XPS analysis performed in the scan mode. Both survey and high-resolution scans were acquired at each profile step, especially to monitor Pb 4f and I 3d spectra. The XPS spectrometer is equipped with a monochromatic Al Kα X-ray beam with a spot size set at 200 μm and the sputtering beam raster area was set to 1 mm to ensure that the analysis was safely performed at the center of the crater. A dual beam flood gun was used for charge compensation. The surface atomic percentages (at.%) were evaluated from survey scans acquired at each profile step and chemical analysis was performed on high-resolution spectra fitted with a Shirley-type background using Avantage© software. 

### 2.4. ToF-SIMS Depth Profile Analysis

ToF-SIMS depth profiles were acquired in non-interlaced mode with the analysis area of 100 × 100 µm^2^ and the raster area of 250 × 250 µm^2^. The profiles with monatomic beams were performed on a ToF-SIMS IV instrument, equipped with a 25 keV Bi_3_^+^ analysis beam, while profiles with Ar_n_^+^ clusters were collected on a ToF-SIMS V instrument equipped with a 30 keV Bi_3_^+^ analysis beam and combined with in-situ AFM (IONTOF GmbH Münster, Germany). The ToF-SIMS sputtering beam conditions are reported in Table 1. Because of the relatively wide cluster size distribution (as shown in Figure 2), we avoided selecting cluster sizes below 500 atoms. Specifically, the cluster size and energy were varied in the range 500 < *n* < 4000 and 5 < E < 20 keV, corresponding to the energy-per-atom range 2.5 eV < E*/n* < 40 eV (see Table 1). A flood gun was used for surface charge compensation and most of the profiles were acquired in both positive and negative polarities. However, only results obtained in negative polarity mode have been exploited because of the higher intensity of characteristic molecular ions signals.

## 3. Results and Discussion

### 3.1. Depth Profiles on Model Perovskite/TiO_2_ Samples

Ion beam induced modifications are first checked on model triple cation perovskite thin films with nominal composition Cs_x_(MA_0.17_FA_0.83_)_100−x_Pb(I_0.83_Br_0.17_)_3_/cTiO_2_/Glass. The most representative XPS depth profiles showing the peak area variation of I 3d, Pb 4f, O 1s, and Cs 3d spectra as a function of the sputtering time are displayed in Figure 3. 

A previously reported, XPS depth profile analysis of CH_3_NH_3_PbI_3_ perovskite solar cells with monatomic beams evidenced the presence of two chemical components in the Pb 4f spectrum. One was ascribed to metallic lead (Pb^0^, with Pb 4f_7/2_ component at ~137 eV binding energy) and the other to lead in the perovskite (Pb^+2^, with Pb 4f_7/2_ component at ~139 eV) [9]. While the origin of such non-negligible Pb^0^ content is still controversial, profiles in Figure 3d–f show that the Pb^0^ content increases with the sputtering time (depth) and it is clearly affected by the ion beam conditions. Namely, when sputtering with Ar_n_^+^ clusters, the Pb^0^ component quickly exceeds 50% of the total Pb 4f peak area, while it represents less than 30% of the total Pb 4f area when sputtering with 1 keV Ar^+^ beam. The lead reduction may possibly occur (*i*) because of the segregation of Pb excess during the perovskite crystallization, leading to the formation of nanoscale Pb^0^ particles, or (*ii*) it may form following the exposure to the X-Rays and/or the ion beam during the depth profile. In both cases, if the Pb^0^ was entirely formed during the depth profile, one would expect a progressive modification of the perovskite surface composition, and possibly, an increase of the I/Pb^+2^ ratio (because of the reduction of Pb^+2^ to Pb^0^). This is not observed, as for almost each ion beam condition, the Pb^0^ increase is associated with a stable or slowly decreasing I/Pb^+2^ ratio (see Figure 3d–f). We conclude that the Pb^0^ is already present in the perovskite layer, however, its content is artificially enhanced by the argon GCIB sputtering because of the slower sputtering of Pb^0^, which tends to accumulate at the sample surface during the profile. 

This explains why the I/Pb ratio (where Pb=Pb^0^+Pb^+2^) decreases faster when profiling with argon clusters with respect to monatomic beams. The Ar_500_^+^ beam results in the most stable I/Pb^+2^ ratio (~2.3, Figure 3f), with a value close to the expected value (the nominal value being I/Pb = 2.5); we conclude that this condition ensures the lowest modification of the perovskite surface composition and the non-accumulation of damages during the profile.

Additional information on the effect of the ion beam sputtering on intrinsically-hybrid perovskites is derived from ToF-SIMS profiles on model samples. The perovskite composition is monitored from the profiles of intense and characteristic molecular ion fragments, including PbI_3_^−^, CH_3_NH_3_I_3_^−^ (MAI_3_^−^), CH_5_N_2_I_2_^−^ (FAI_2_^−^), CN^−^ and I_2_^−^. The molecular profiles obtained with low-energy monatomic sputtering beams (Ar^+^ and Cs^+^) are displayed in Figure 4. Compared to Ar^+^, (Figure 4a,d and Appendix A), Cs^+^ profiles (Figure 4b–f) result in clearly less-variable and more intense molecular signals. The signal enhancement is related to the surface implantation of Cs atoms which increase the ionization yield of the negatively-charged sputtered ions [33,34]. To evaluate and compare the ion beam induced modifications between the different sputtering conditions, we have selected two ratio indicators for the fragmentation of organic molecules (CN^−^/FAI_2_^−^) and the preferential sputtering of inorganic species with respect to organic ones (PbI_3_^−^/CN^−^). 

Generally, flat molecular profiles (signals or ratios) would indicate that the ion beam induced modifications do not accumulate during the profile. The 500 eV Cs^+^ sputtering results in the least-variable indicator profiles and ensures the high sputtering rate (~2.5 nm/s) in the perovskite layer. Monatomic Ar^+^ results in highest sputtering rate but also in more variable intensity ratios as compared to Cs^+^ (Figure 4d). Moreover, with respect to Ar^+^, the fragmentation indicator value is roughly halved with Cs^+^ sputtering, possibly because of the increase of the ionization yield of the FAI_2_^−^ fragment.

ToF-SIMS depth profiles on model hybrid samples have been carried with variable-size argon clusters (Ar*_n_*^+^) with energy from 5 to 20 keV. At a constant energy of E = 10 keV, the cluster size sensibly impacts the sputtering rate; in particular, the sputtering rate obtained on the hybrid perovskite layer is ~1 nm/s with Ar_500_^+^ (Figure 5), ~0.3 nm/s with Ar_1000_^+^, and ~0.15 nm/s with Ar_4000_^+^ (Appendix A). At energies above 10 keV, the argon cluster size has a modest impact on the fragmentation indicator (see Appendix A, and Figure 5e), while the preferential sputtering of organic molecules increases dramatically when increasing the cluster size above *n* = 1000 (Appendix A).

Based on these results, the optimum argon cluster size is found to be a few hundred atoms (*n* = 500). For this size (Ar_500_^+^), we systematically investigated the effect of the cluster energy (Figure 5). At 5 keV (*E/n* = 10 eV), the Ar_500_^+^ sputter rate is only ~0.12 nm/s, the perovskite/cTiO_2_ interface is broad, and molecular signals tend to decrease during the profile. In these conditions, the positive surface charging generated by the ion sputtering could possibly promote the migration of negatively charged ion species to the crater surface. This phenomenon can explain the progressive iodine decrease during the profile (Figure 5a,b), however, it does not apply to Br^−^ signal, which rather tends to accumulate toward the bottom interface during long profiles. We conclude that the charging-related ion migration is ruled by the specific ion mobility into the perovskite [35]. 

With the 20 keV Ar_500_^+^ beam, the cTiO_2_ interface is reached after only 100 s, corresponding to a sputtering rate of the perovskite material of ~5 nm/s, as confirmed by the AFM profile of the crater region acquired in-situ, showing a 200 nm depth after 40 s sputtering (see Appendix A). The 20 keV Ar_500_^+^ profile features constant molecular ions (and indicators) profiles in the whole perovskite layer depth (Figure 5f), indicating that ion beam induced damages do not accumulate during the profile and no preferential sputtering occurs on the different perovskite constituents. Moreover, the fragmentation indicator value is halved with respect to monatomic Ar^+^ (Figure 4d), and the Br^−^ signal does not increase at the perovskite/cTiO_2_ interface.

The effect of the sputtering beam on the hybrid perovskite surface was further investigated by acquiring in-situ AFM topography images inside the crater region during the profile (Figure 6). The initial topography of the perovskite layer features µm-sized monolithic perovskite grains (Appendix A). Clearly, the sputtering affects the surface topography; specifically, while argon clusters with *n* = 1000 and E*/n* = 10 eV (see Figure 6a and the corresponding profile in Appendix A) tend to preserve the initial morphology and grains are still recognizable in Figure 6a, argon clusters with smaller size (*n* = 500) or higher energy (*E/n* = 40 eV) progressively deform the surface topography (Figure 6b,c). The progressive amorphization of the crater surface is confirmed by the lowering of the surface roughness and the disappearance of the grain edges for the crater obtained with Ar_500_^+^ at *E/n* = 40 eV (Figure 6c).

In a simplified view, we can sum up that for argon clusters sizes above *n* = 1000 or cluster energies below 10 keV (or *E/n* < 20 eV), the sputtering rate drops and the molecular signals profiles (and their intensity ratios) tend to artificially evolve with the sputtering time. This result agrees with XPS depth profiles showing that large clusters do not efficiently sputter metallic lead. Overall, ToF-SIMS profiles on model samples indicate that the most efficient sources to profile hybrid perovskite are low-energy Cs^+^ or argon clusters of a few hundred atoms with energy above 10 keV. These conditions ensure high sputtering rates (few nm/s) and rather constant molecular profiles. In these conditions, the defects induced by both the analysis and the sputtering beams are efficiently removed during the subsequent erosion cycle. 

### 3.2. Depth Profiles on Full Perovskite Solar Cells

The optimum depth profile conditions resulting from the analysis of model samples have been applied to mesoscopic PSCs with structure Au/spiro-OMeTAD/(FA_x_Cs_1−x_PbI_3_)_0.85_(MAPbBr_3_)_0.15_/m-TiO_2_/cTiO_2_/FTO/Glass (Figure 1). Since the maximum cluster energy available on our XPS spectrometer is 8 keV and in order to reduce the profile duration, the top Au electrode and part of the spiro-OMeTAD layer were previously sputtered with the 1 keV Ar^+^ beam, and after that, the ion source was switched to the cluster mode (at 8 keV). XPS profiles obtained with Ar^+^ (1 keV), Ar_75_^+^ and Ar_500_^+^ are shown in Figure 7. For Ar_75_^+^, despite the high energy-per-atom value (E*/n*~100 eV), the sputtering rate is only ~0.1 nm/s, and the I/Pb ratio decreases similarly to the Ar_500_^+^ profile. This suggests that the sputtering of Pb^0^ particles is also rather ineffective with small clusters, as confirmed by the rather slow decay of the residual Au signal (Figure 7b).

In the explored energy range, argon clusters result in the Pb^0^/Pb ratio quickly exceeding 50%; this explains the fast decay of I/Pb. However, the constant I/Pb^+2^ ratio observed for the cluster profiles indicate that surface modifications of the perovskite stoichiometry do not accumulate during the profile. The I/Pb^+2^ value in the perovskite layer is closer to the nominal value 2.5 with Ar_500_^+^ sputtering, while a lower iodine content (I/Pb^+2^~2) is measured at the perovskite surface is with Ar^+^ and Ar_75_^+^ (the XPS analysis refers to the first 10 nm depth). 

ToF-SIMS depth profiles obtained on full mesoscopic PSCs are shown in Figure 8. For 1 keV Ar^+^ sputtering (Figure 8a), the Au and spiro-OMeTAD layers are efficiently profiled in only ~500 s. The monatomic argon profile results in well-defined interfaces, suggesting a rather low intermixing or roughening at hybrid interfaces. However, signals from organic molecular fragments are rather weak (FAI_2_^−^), suggesting a high degree of fragmentation in the top surface.

For 500 eV Cs^+^ sputtering (Figure 8b), the PSC profile shows sharp interfaces and more intense organic and inorganic molecular signals thanks to the enhanced negative ionization promoted by Cs atoms implantation. Finally, for 20 keV Ar_500_^+^ (Figure 8c), on the one hand, the sputtering time to profile the top Au electrode is about ten times higher (~2000 s) with respect to 1 keV Ar^+^. On the other hand, the sputtering of the perovskite layer and the m-TiO_2_ are much more efficient, resulting in sharp interfaces. The rather slow drop of the Au_3_^-^ signal at the Au/spiro-OMeTAD interface confirms the inefficient sputtering of small metal particles; this fact is of interest when investigating the gold diffusion in aged devices.

## 4. Conclusions

Model triple-cation perovskite thin layers and mesoscopic PSCs are depth profiled with different sputtering ion beam conditions including low-energy monatomic ion beams (Ar^+^ and Cs^+^) to argon GCIBs at different sizes (*n*) and energies (*E, E/n*). For each condition, the main ion beam induced modifications are assessed by evaluating the fragmentation, preferential sputtering, chemical modifications, and the interface sharpness (roughness). Overall, sufficiently high erosion rates (nm/s) are found to be beneficial in preventing the accumulation of damages and in limiting the total ion dose during the profile (from the analysis and sputtering beams in ToF-SIMS). 

Compared to low-energy Ar^+^, Ar_500_^+^ clusters allow us to sensibly reduce the fragmentation of organic molecules thanks to the lower energy-per-atom and the lower implantation depth. However, non-optimized cluster beams result in the preferential sputtering of organic molecules and are inefficient on metals, in particular for the sputtering of small metal clusters (Au, Pb^0^). This limitation can be attenuated by choosing few-hundred-atoms clusters and by increasing the energy above 10 keV. In the explored range, the best conditions to profile hybrid perovskites are with 500 eV Cs^+^ and Ar_500_^+^ at 20 keV. In these conditions, we obtain extremely stable ToF-SIMS molecular profiles, stable ratio indicators, and high sputtering rates (few nm/s). In ToF-SIMS profiles, low-energy Cs^+^ sputtering sensibly increases the negative ionization yields. 

While Ar^+^ provides for highest sputtering rate, XPS depth profiles have shown that the surface stoichiometry of the perovskite (I/Pb^+2^) is slightly altered by the Ar^+^ beam, which also strongly reduces the Ti atoms from TiO_2_ by the preferential sputtering of oxygen. The best conditions have been successfully applied to depth profile PSCs, resulting in well-defined interfaces and intense molecular signals. These results are believed to be helpful in guiding the choice of the sputtering beam conditions in a wider range of hybrid materials.

## Figures and Tables

**Figure 1 materials-12-00726-f001:**
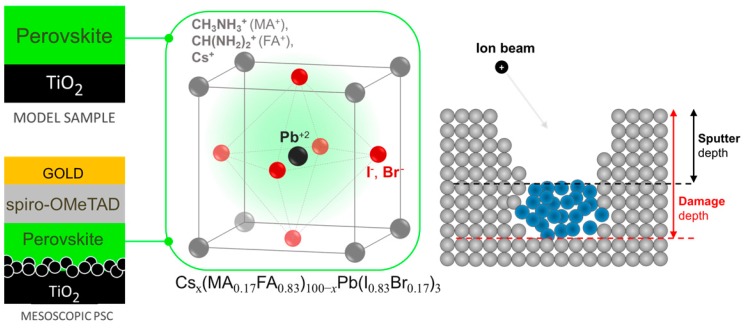
Hybrid perovskite sample schemes and stack composition. The strategy is to identify the best sputtering beam conditions on a model sample and then apply them to profile mesoscopic perovskite solar cells. The basic requirement to avoid the accumulation of damage is that, for each sputtering/analysis cycle, the sputtered depth, as defined in the right scheme, must exceed half of the damage depth induced by the sputtering and the analysis beams.

**Figure 2 materials-12-00726-f002:**
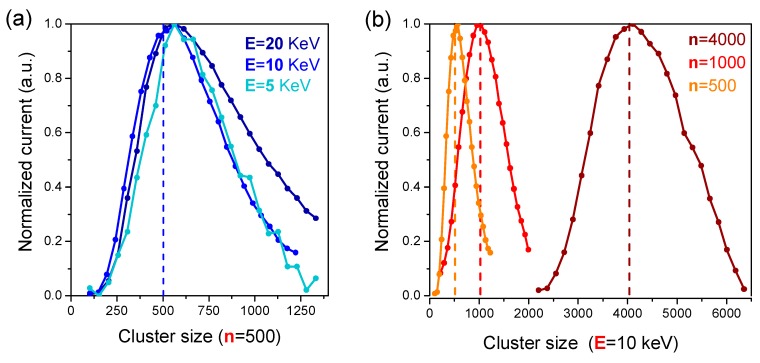
Size distribution (*n*) of Ar_n_^+^ clusters on the IONTOF V ToF-SIMS spectrometer at (**a**) constant nominal size (*n* = 500 as indicated by the vertical line) and variable cluster energy (5 < E < 20 keV), and (**b**) fixed energy (E = 10 keV) and variable nominal cluster size (500 < *n* < 4000).

**Figure 3 materials-12-00726-f003:**
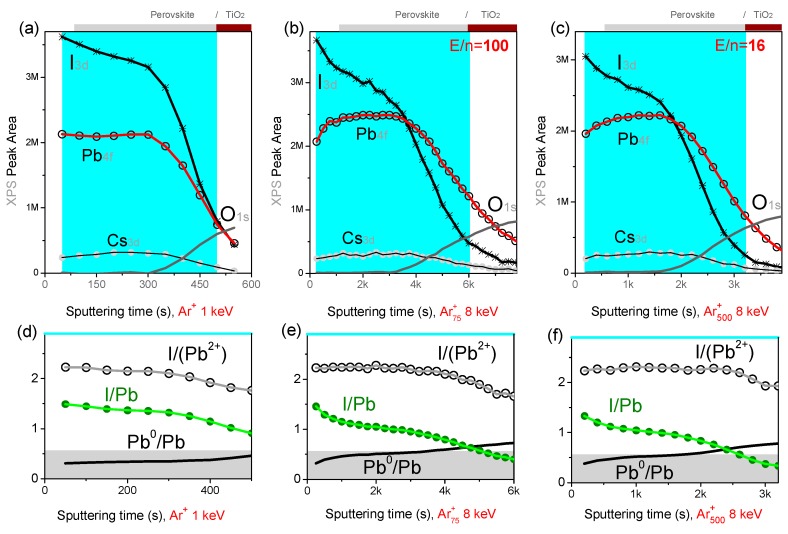
XPS depth profiles acquired on Perovskite/cTiO2 model samples obtained with (**a**,**d**) monoatomic Ar^+^ at 1 keV; (**b**,**e**) Ar_75_^+^ clusters at 8 keV; and (**c**,**f**) Ar_500_^+^ clusters at 8 keV. Bottom panels show atomic percent ratios between iodide and lead which are present in two different chemical states (Pb^0^ and Pb^+2^ from the perovskite). Bottom panels refer to the perovskite layer highlighted in the top panels for which the upper limit corresponds to the O 1s signal (from cTiO_2_) reaching 70% of its plateau value.

**Figure 4 materials-12-00726-f004:**
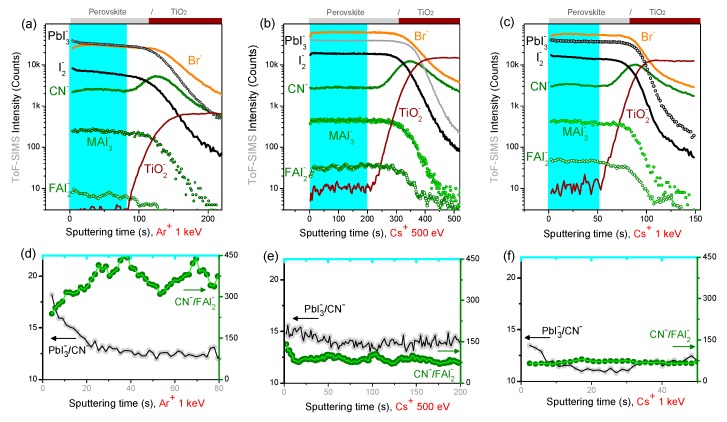
ToF-SIMS depth profiles acquired on Perovskite/cTiO_2_ model samples obtained with monoatomic beams. (**a**,**d**): Ar^+^ at 1 keV; (**b**,**e**): Cs^+^ at 500 eV; (**c**,**f**): Cs^+^ at 1 keV. Bottom panels show a selected indicator for the preferential sputtering of inorganic/organic species (PbI_3_^−^/CN^−^) and an indicator for molecular fragmentation (CN^−^/FAI_2_^−^) in the highlighted perovskite region, i.e., before the rise of cTiO_2_^−^ signal. Constant profiles indicate the non-accumulation of damages.

**Figure 5 materials-12-00726-f005:**
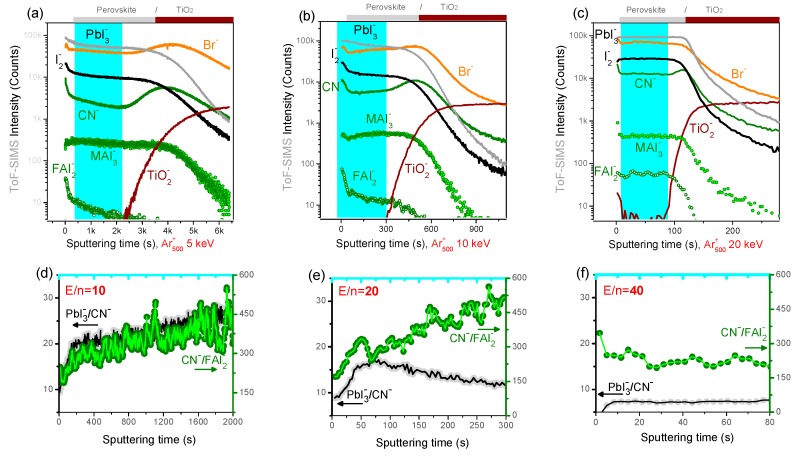
ToF-SIMS depth profiles acquired on Perovskite/cTiO_2_ model samples with constant argon cluster size (*n* = 500) and variable energy. Constant profiles indicate the non-accumulation of damages during the profile. (**a**,**d**): E = 5 keV (*E/n* = 10 eV); (**b**,**e**): E = 10 keV (*E/n* = 20 eV); (**c**,**f**): E = 20 keV (*E/n* = 40 eV). Bottom panels show that (*i*) a higher preferential sputtering of organic molecules occurs at low cluster energy (PbI_3_^-^/CN^-^ ratios) and (*ii*) a lower fragmentation of organic molecules occurs with 20 keV Ar_500_^+^ beam (CN^-^/FAI_2_^-^).

**Figure 6 materials-12-00726-f006:**

In-situ AFM images acquired in the sputtered crater after 250 s (~210 nm depth) of Ar_1000_^+^ sputtering at 10 keV (**a**), 750 s (~160 nm depth) of Ar_500_^+^ sputtering at 5 keV (**b**), and 40 s (~220 nm depth) of Ar_500_^+^ sputtering at 20 keV (**c**). Monolithic perovskite grains are still visible in (**a**) and progressively disappear in (**c**).

**Figure 7 materials-12-00726-f007:**
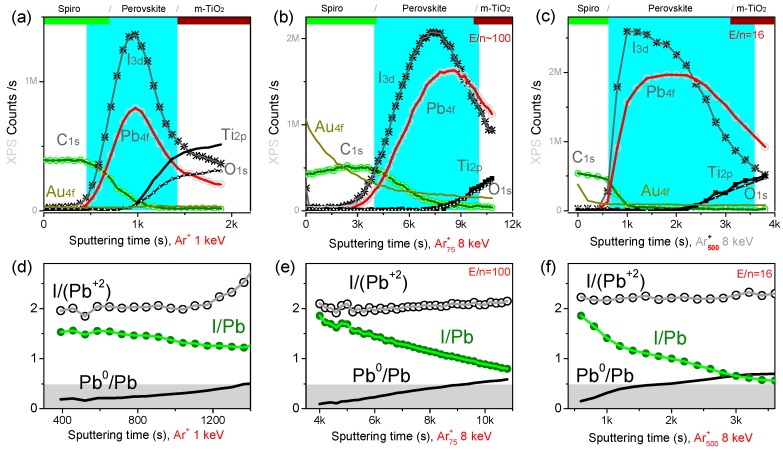
XPS depth profiles acquired on PSCs obtained with (**a**,**d**) monoatomic Ar^+^ at 1 keV, (**b**,**e**) Ar_75_^+^ clusters at 8 keV, and (**c**,**f**) Ar_500_^+^ clusters at 8 keV. Bottom panels show atomic percent rations between iodide and lead which are present in two different chemical states (metallic Pb^0^ and Pb^+2^ from the perovskite) in the perovskite area. The perovskite layer limits are roughly identified by the rise of the iodine and oxygen signals.

**Figure 8 materials-12-00726-f008:**
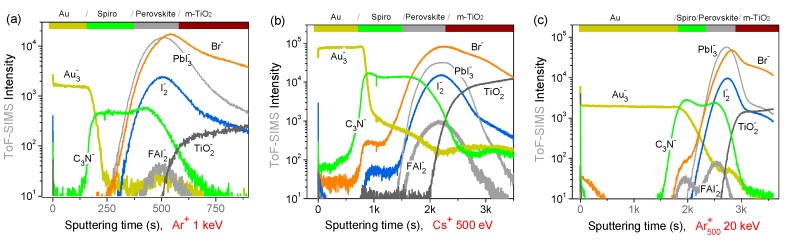
ToF-SIMS depth profiles of PSCs with (**a**) 1 keV Ar^+^, (**b**) 500 eV Cs^+^, and (**c**) 20 keV Ar_500_^+^.

**Table 1 materials-12-00726-t001:** Ion beam sputtering conditions: Cluster size, energy, energy-per-atom, and typical current values in ToF-SIMS depth profiles.

Sputtering Beam	E (keV)	E/*n* (eV)	Ion Current (nA)
Ar_4000_^+^	10	2.5	1
Ar_1000_^+^	10, 20	10, 20	0.5
Ar_500_^+^	5, 10, 20	10, 20, 40	0.5
Ar^+^	0.5, 1	/	~100
Cs^+^	0.5, 1	/	35, 75

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
