# Peer review of "Hybrid Perovskites Depth Profiling with Variable-Size Argon Clusters and Monatomic Ions Beams"

_materials, 2019, doi:10.3390/ma12050726_

Round 1

Reviewer 1 Report

In this manuscript, authors prepared triple-cation perovskite thin layers and solar cells in order to obtain their depth profile under various sputtering source conditions, such as low-energy monatomic ion beans and Ar GCIBs. The damages caused by sputtering during device fabrication were analyzed by TOF-SIMS, XPS AND AFM. By comparing various sputtering conditions and their experimental results, the authors also provide their advantages and disadvantages, which can hopefully be used as a guide to choose sputtering sources/conditions in wide range of hybrid materials. The manuscript is interesting, however, there are some points that need to be addressed before publication. 

1. Please check your manuscript for typos and grammar mistakes, e.g. ln 40, thin films stacks?; ln 134, lead compoounds from TCI?

2. Please provide clear version for Fig 6. They are blurred.

3. Please add data of the device power conversion efficiency using various sputtering sources/conditions during device fabication, which can improve the quality of the whole manuscript, and provide a more straight-forward way of understanding the sputtering choices.

Author Response

In this manuscript, authors prepared triple-cation perovskite thin layers and solar cells in order to obtain their depth profile under various sputtering source conditions, such as low-energy monatomic ion beans and Ar GCIBs. The damages caused by sputtering during device fabrication were analyzed by TOF-SIMS, XPS AND AFM. By comparing various sputtering conditions and their experimental results, the authors also provide their advantages and disadvantages, which can hopefully be used as a guide to choose sputtering sources/conditions in wide range of hybrid materials. The manuscript is interesting; however, there are some points that need to be addressed before publication.

We would like to point out the fact that in our work, the ion beam sputtering was always performed after (i.e. not during) the device fabrication.

Point 1: Please check your manuscript for typos and grammar mistakes, e.g. ln 40, thin films stacks? ln 134, lead compounds from TCI?
Response 1: We have checked for typo errors and performed the suggested corrections (thin film stacks, TCI chemicals and some other minor grammatical mistakes)

Point 2: Please provide clear version for Fig 6. They are blurred.
Response 2: As suggested by the reviewer, we have uploaded a higher resolution image; the overall quality of the image now depends on the original AFM file image and not on the image resolution.

Point 3: Please add data of the device power conversion efficiency using various sputtering sources/conditions during device fabrication, which can improve the quality of the whole manuscript, and provide a more straightforward way of understanding the sputtering choices.

Response 3: We believe that the reviewer had a misunderstanding of our experimental setup;
To be clear, the different ion beam sputtering conditions were used to characterize the devices after (and not during) the device processing. Therefore, the Power Conversion Efficiency has not been investigated in this work and the suggested discussion does not seem to us to apply because the analysis is destructive.

Reviewer 2 Report

The manuscript by C. Noël et al. presents depth profiling of hybrid perovskite layers using low energy monoatomic beams and variable-size Ar clusters. The authors optimize the best ion beam conditions by systematically comparing the sputtering rates and the surface modifications associated with sputtering beam. The authors demonstrate the use of techniques such as XPS, ToF-SIMS, and SPM to characterize the various perovskite interfaces. With monoatomic beams, 500 eV Cs+ shows best results to characterize the profile of hybrid perovskites. Among variable-size Ar clusters, few hundred argon atoms (300 < n < 500) and E/n ~ 20 eV has been shown to be effective for profiling of hybrid perovskites. The manuscript is well written and the design of experiments is solid. The results presented in the manuscript would benefit the readership of Materials journal. I would recommend to ACCEPT the manuscript for publication after the following minor concerns of the reviewer are addressed:

1.       Page 2, line 92, it should be ‘result’ and not ‘results’.

2.       Page 4, line 132, wouldn’t the perovskite layer be deposited on c-TiO2 rather than FTO?

3.       Page 4, line 178, for the given parameters, would the range of energy per atom be 1.25 < E/n < 40? Also, the authors should include the units (eV) here.

Author Response

The manuscript by C. Noël et al. presents depth profiling of hybrid perovskite layers using low energy monoatomic beams and variable-size Ar clusters. The authors optimize the best ion beam conditions by systematically comparing the sputtering rates and the surface modifications associated with sputtering beam. The authors demonstrate the use of techniques such as XPS, ToF-SIMS, and SPM to characterize the various perovskite interfaces. With monoatomic beams, 500 eV Cs+ shows best results to characterize the profile of hybrid perovskites. Among variable-size Ar clusters, few hundred argon atoms (300 < n < 500) and E/n ~ 20 eV has been shown to be effective for profiling of hybrid perovskites. The manuscript is well written and the design of experiments is solid. The results presented in the manuscript would benefit the readership of Materials journal. I would recommend to ACCEPT the manuscript for publication after the following minor concerns of the reviewer are addressed:

Point 1: Page 2, line 92, it should be ‘result’ and not ‘results’.
Response 1
: Thank you, the mistake was corrected in the text.

Point 2: Page 4, line 132, wouldn’t the perovskite layer be deposited on c-TiO2 rather than FTO?
Response 2:
Indeed, the perovskite layer was deposited on c-TiO2, which itself is deposited on FTO coated glass. The sentence was corrected accordingly.

Point 3: Page 4, line 178, for the given parameters, would the range of energy per atom be 1.25 < E/n < 40? Also, the authors should include the units (eV) here.
Response 3: The lower energy used at the larger cluster size (n = 4000) was 10 keV. Therefore, the lower E/n is 2.5 eV. In order to clarify the message, we refer to Table 1 in the text. Units for E/n were added through all the revised MS.
